# Identification of Two Novel R2R3-MYB Transcription factors, *PsMYB114L* and *PsMYB12L*, Related to Anthocyanin Biosynthesis in *Paeonia suffruticosa*

**DOI:** 10.3390/ijms20051055

**Published:** 2019-02-28

**Authors:** Xinpeng Zhang, Zongda Xu, Xiaoyan Yu, Lanyong Zhao, Mingyuan Zhao, Xu Han, Shuai Qi

**Affiliations:** 1State Key Laboratory of Crop Biology, College of Horticulture Science and Engineering, Shandong Agricultural University, Taian 271018, China; zhxpnd123@163.com; 2College of Forestry, Shandong Agricultural University, Taian 271018, China; xuzoda@163.com (Z.X.); zhaomingy9@163.com (M.Z.); hanxusdau@163.com (X.H.); shuaiqi@sdau.edu.cn (S.Q.)

**Keywords:** *P. suffruticosa*, R2R3-MYB, overexpression, anthocyanin, transcriptional regulation

## Abstract

Flower color is a charming phenotype with very important ornamental and commercial values. Anthocyanins play a critical role in determining flower color pattern formation, and their biosynthesis is typically regulated by R2R3-MYB transcription factors (TFs). *Paeonia suffruticosa* is a famous ornamental plant with colorful flowers. However, little is known about the R2R3-MYB TFs that regulate anthocyanin accumulation in *P. suffruticosa*. In the present study, two R2R3-MYB TFs, namely, *PsMYB114L* and *PsMYB12L*, were isolated from the petals of *P. suffruticosa* ‘Shima Nishiki’ and functionally characterized. Sequence analysis suggested that *PsMYB114L* contained a bHLH-interaction motif, whereas *PsMYB12L* contained two flavonol-specific motifs (SG7 and SG7-2). Subsequently, the in vivo function of *PsMYB114L* and *PsMYB12L* was investigated by their heterologous expression in *Arabidopsis thaliana* and apple calli. In transgenic *Arabidopsis* plants, overexpression of *PsMYB114L* and of *PsMYB12L* caused a significantly higher accumulation of anthocyanins, resulting in purple-red leaves. Transgenic apple calli overexpressing *PsMYB114L* and *PsMYB12L* also significantly enhanced the anthocyanins content and resulted in a change in the callus color to red. Meanwhile, gene expression analysis in *A. thaliana* and apple calli suggested that the expression levels of the flavonol synthase (*MdFLS*) and anthocyanidin reductase (*MdANR*) genes were significantly downregulated and the dihydroflavonol 4-reductase (*AtDFR*) and anthocyanin synthase (*AtANS*) genes were significantly upregulated in transgenic lines of *PsMYB114L*. Moreover, the expression level of the *FLS* gene (*MdFLS*) was significantly downregulated and the *DFR* (*AtDFR*/*MdDFR*) and *ANS* (*AtANS*/*MdANS*) genes were all significantly upregulated in transgenic lines plants of *PsMYB12L*. These results indicate that *PsMYB114L* and *PsMYB12L* both enhance anthocyanin accumulation by specifically regulating the expression of some anthocyanin biosynthesis-related genes in different plant species. Together, these results provide a valuable resource with which to further study the regulatory mechanism of anthocyanin biosynthesis in *P. suffruticosa* and for the breeding of tree peony cultivars with novel and charming flower colors.

## 1. Introduction

*Paeonia suffruticosa* is a very popular ornamental flowering plant that was first cultivated more than 1600 years ago in China and is currently distributed worldwide. This species is in the Paeoniaceae family and has been named ‘the king of flowers’ for its showy and colorful flowers [1]. Approximately 1500 cultivars of *P. suffruticosa* with a variety of flower colors have been produced by breeders worldwide [2]. Among the many flower colors of this species, most fit into two clusters: monochrome color (red, pink, white, purple, black, blue, green, and yellow) and double color. Cultivars with a double-color phenotype are rarer and more sought after, and thus have great ornamental and commercial value [3]. Among them, *P. suffruticosa* ‘Shima Nishiki’, a well-known chimeric cultivar, was selected from the bud mutation of *P. suffruticosa* ‘Taiyoh’. ‘Shima Nishiki’ usually has red and pink petals on the same flower, and this aesthetically pleasing double-color phenotype can be stably inherited [4]. Therefore, the ‘Shima Nishiki’ cultivar is regarded as an important experimental material with which to study the molecular regulatory mechanism of flower color and in the breeding of new cultivars [5].

Anthocyanins are important soluble flavonoid compounds that are widely distributed in the leaves, flowers, fruits, seeds and other tissues of many plants [6]. Anthocyanin composition and concentration are usually closely related to flower color intensity [7,8]. The anthocyanin biosynthetic pathway is well known to be highly conserved in many ornamental plants [9,10,11,12,13,14]. Anthocyanin biosynthesis and accumulation are usually regulated by a series of structural genes and regulatory genes [15,16]. The structural genes encode enzymes associated with anthocyanin biosynthesis, including chalcone synthase (CHS), chalcone isomerase (CHI), flavanone 3-hydroxylase (F3H), flavonoid 3′-hydroxylase (F3′H), dihydroflavonol 4-reductase (DFR) anthocyanin synthase (ANS), Flavonol synthase (FLS), and anthocyanidin reductase (ANR) [17,18,19] (Figure 1). Among them, FLS is a dedicated enzyme involved in flavonol biosynthesis, and ANR is a key enzyme for proanthocyanidin biosynthesis. The regulatory genes can be divided into three families R2R3-MYB, bHLH, and WD40 [20,21,22] and they usually form a regulatory complex to activate the expression of anthocyanin biosynthetic genes [23,24,25].

Many structural genes have been characterized and cloned in *P. suffruticosa* [12,26,27,28]. In the MYB-bHLH-WDR (MBW) complex, R2R3-MYB transcription factors (TFs) usually play critical roles in anthocyanin biosynthesis and accumulation [29,30]. Many R2R3-MYB TFs involved in anthocyanin biosynthesis have been isolated and characterized from various plants, including *Arabidopsis thaliana* [24], *Zea mays* [31], *Vitis vinifera* [32], *Malus* crabapple [33], *Petunia hybrida* [34], *Antirrhinum majus* [29], *Dendranthema morifolium* [35] and *Phalaenopsis aphrodite* [14]. In *P. suffruticosa*, most previous studies were focused primarily on the preliminary investigation of R2R3-MYB TFs based on transcriptome sequencing and qRT-PCR analyses [12,36,37,38], whereas whether and how R2R3-MYB TFs control anthocyanin biosynthesis and accumulation in *P. suffruticosa* are almost unknown.

In the present study, two novel R2R3-MYB TFs, namely, *PsMYB114L* and *PsMYB12L*, were cloned in *P. suffruticosa*. Subsequently, the expression patterns of *PsMYB114L* and *PsMYB12L* were determined at five developmental stages in *P. suffruticosa* ‘Shima Nishiki’. Furthermore, the function of these two TFs was further verified by heterologous expression in *Arabidopsis* and apple calli. These results will provide valuable insights into understanding the putative roles of *PsMYB114L* and *PsMYB12L* in regulating anthocyanin biosynthesis in *P. suffruticosa.*

## 2. Results

### 2.1. Cloning and Analysis of the PsMYB114L and PsMYB12L Genes

Based on the functional annotation and gene expression analysis of transcriptome sequencing data in *P. suffruticosa* ‘Shima Nishiki’ [39], we filtered two MYB unigenes exhibiting relatively high expression differences between the red and pink petals as the targeted genes of this study.

The full-length cDNA sequences of the two novel MYB genes were obtained with PCR amplification. By conducting GenBank BLAST searches of the amino acid sequences of these two genes, we found that these genes have the highest homology with transcription factor *MYB114*-like [*Quercus suber*] and transcription factor *MYB12*-like [*Juglans regia*], respectively. Therefore, we named these genes *PsMYB114L* and *PsMYB12L*. Sequencing results revealed that *PsMYB114L* (Appendix A) and *PsMYB12L* (Appendix A) contained an open reading frame (ORF) of 600 and 1140 bp encoding 199 and 379 amino acids and that their predicted proteins had a molecular mass of 22.81 and 42.61 kDa and a theoretical isoelectric point (pI) of 8.53 and 4.86, respectively.

Multiple sequence alignment of amino acids revealed that *PsMYB114L* and *PsMYB12L*, belonging to the SANT superfamily (which typically consists of tandem repeats of three alpha-helices arranged in a helix-turn-helix motif, with each alpha helix containing a bulky aromatic residue), and other known R2R3-MYB TFs related to anthocyanin biosynthesis contained a highly conserved R2R3 DNA-binding domain. The presence of this conserved domain means that *PsMYB114L* and *PsMYB12L* are also R2R3-MYB TFs and may perform similar functions in regulating anthocyanin biosynthesis. Furthermore, *PsMYB114L* had a bHLH-interaction motif ([D/E]Lx2[R/K]x3Lx6Lx3R) in the R3 domain at the N terminus and did not have any conserved motifs at the C terminus (Figure 2A). Moreover, *PsMYB12L* did not have any bHLH-interaction motifs at the N terminus, whereas it contained two flavonol-specific motifs [40], namely, SG7 ([K/R][R/x][R/K]xGRT[S/x][R/G]xx[M/x]K) and SG7-2 ([W/x][L/x]LS), at the C terminus (Figure 2B).

To better evaluate the phylogenetic relationships of *PsMYB114L*, *PsMYB12L* and 16 other known MYB TFs related to the regulation of anthocyanin biosynthesis, a phylogenetic tree was constructed based on the amino acid sequences of these 18 MYB TFs from different species using the neighbor-joining method. The phylogenetic analysis indicated that these 18 MYB TFs were classified into four groups (Flavonol, Anthocyanin, Anthocyanin/Proanthocyanidin and Proanthocyanidin) based on their specific roles in the flavonoid biosynthesis pathway. Among them, *PsMYB114L* had the closest phylogenetic relationship with *ZmC1* and *ZmPL*, which are involved in regulating anthocyanin biosynthesis, whereas *PsMYB12L* belongs to a subgroup of MYB proteins that includes VvMYBF1, EsMYBF1, AtMYB11, AtMYB12 and AtMYB111, which regulate flavonol synthesis and had the closest phylogenetic relationship with *VvMYBF1* (Figure 3).

### 2.2. Subcellular Localization of PsMYB114L and PsMYB12L

To examine the subcellular localization of *PsMYB114L* and *PsMYB12L*, the recombinant vector (*PsMYB114L*-GFP/*PsMYB12L*-GFP) and the control vector (pCAMBIA1301-GFP) were introduced into the tobacco leaves. Our results were basically consistent with those of previous studies [41,42]. The green fluorescent protein (GFP) fluorescence of the control vector was clearly distributed throughout the entire cell (Figure 4A), and the *PsMYB114L*-GFP/*PsMYB12L*-GFP vector displayed a strong fluorescence signal in the nucleus and cytoplasm of tobacco cells (Figure 4B,C). Therefore, we speculated that the two R2R3-MYB TFs (*PsMYB114L*/*PsMYB12L*) were simultaneously localized and functioned in the nucleus and cytoplasm.

### 2.3. Expression Patterns of PsMYB114L and PsMYB12L in P. suffruticosa ‘Shima Nishiki’

qRT-PCR analysis was conducted to survey the expression patterns of *PsMYB114L* and *PsMYB12L* in *P. suffruticosa* ‘Shima Nishiki’ (Figure 5). Petal samples of this cultivar were collected at five developmental stages (Appendix A). The expression levels of the *PsMYB114L* gene peaked at S3 and then decreased from S3 to S5, whereas the *PsMYB12L* gene exhibited the highest expression at S4. Furthermore, the expression levels of the eight anthocyanin biosynthesis-related genes (*PsCHS*, *PsCHI*, *PsF3H*, *PsF3′H*, *PsDFR*, *PsANS*, *PsFLS*, and *PsANR*) were analyzed. Among these genes, *PsF3′H*, *PsDFR*, and *PsANS* showed a trend similar to that of *PsMYB12L*, whereas *PsFLS* and *PsANR* showed a trend similar to that of *PsMYB114L*.

### 2.4. Overexpression of PsMYB114L and PsMYB12L in Arabidopsis

To characterize the functions of *PsMYB114L* and *PsMYB12L*, these two genes under the expression of the 35S promoter were genetically transformed into *Arabidopsis*. Phenotypic investigations of the transgenic lines of *PsMYB114L* and *PsMYB12L* revealed that their leaves were much deeper in color than those of Col-0 and showed a purple-red color (Figure 6A). Meanwhile, these transgenic lines of the two genes were confirmed by PCR analysis (Figure 6B). Furthermore, the total anthocyanin content results indicated that the transgenic lines of *PsMYB114L* and *PsMYB12L* produced much more anthocyanin than Col-0 (Figure 6C,D).

Additionally, the expression levels of anthocyanin biosynthesis-related genes (*AtCHS*, *AtCHI*, *AtF3H*, *AtF3′H*, *AtDFR*, *AtANS*, *AtFLS*, and *AtANR*) in the Col-0 and the transgenic *Arabidopsis* plants of *PsMYB114L* and *PsMYB12L* were analyzed with qRT-PCR experiments. Compared with the Col-0, overexpression of *PsMYB114L* upregulated the expression of most of the genes (*AtCHS*, *AtCHI*, *AtF3H*, *AtF3′H*, *AtDFR*, and *AtANS*) in transgenic *PsMYB114L* plants; among them, both of the *AtDFR*/*AtANS* genes showed a relatively high difference between the Col-0 and transgenic plants, whereas *AtFLS* and *AtANR* were downregulated in transgenic *PsMYB114L* plants (Figure 6E).

For *PsMYB12L* overexpression in *Arabidopsis*, the expression levels of all eight genes were upregulated in transgenic *PsMYB12L* plants. Among them, the four genes (*AtCHS*, *AtCHI*, *AtDFR*, and *AtANS*) all showed a relatively high difference between the Col-0 and transgenic plants (Figure 6F).

### 2.5. Overexpression of PsMYB114L and PsMYB12L in Apple Calli

For further functional validation, the two genes (*PsMYB114L* and *PsMYB12L*) were ectopically expressed in the calli of ‘Orin’ apple. Interestingly, after light and low-temperature treatments, the WT had almost no phenotypic changes, but an especially obvious color change was observed in the transgenic lines of *PsMYB114L* and *PsMYB12L* (Figure 7A and Figure 8A). PCR amplification confirmed that these transgenic apple calli carry *PsMYB114L* and *PsMYB12L* (Figure 7B and Figure 8B). With regard to the total anthocyanin content, the transgenic lines of *PsMYB114L* and *PsMYB12L* all accumulated markedly higher amounts of anthocyanins than did the WT (Figure 7C,D and Figure 8C,D).

Additionally, the expression levels of anthocyanin biosynthesis-related genes (*MdCHS*, *MdCHI*, *MdF3H*, *MdF3′H*, *MdDFR*, *MdANS*, *MdFLS*, and *MdANR*) in the WT and the transgenic lines of *PsMYB114L* and *PsMYB12L* were analyzed by qRT-PCR. Compared with the WT, overexpression of *PsMYB114L* downregulated the expression of most of the genes, specifically, *MdCHS*, *MdCHI*, *MdF3H*, *MdF3′H*, *MdFLS*, and *MdANR*, and upregulated the expression of *MdDFR* and *MdANS* in transgenic *PsMYB114L* calli (Figure 7E).

For *PsMYB12L* overexpression, the expression levels of most genes, including *MdCHS*, *MdF3H*, *MdF3′H*, *MdDFR*, *MdANS*, and *MdANR*, were upregulated, but those of *MdCHI* and *MdFLS* were downregulated in transgenic *PsMYB12L* calli (Figure 8E).

## 3. Discussion

Flower color is a very important trait in many ornamental plants and has a close association with their ornamental and commercial value. Many prior studies have shown that anthocyanins are a key factor influencing flower color [43,44,45]. R2R3-MYB TFs comprise one of the largest gene families in plants and play key roles in regulating anthocyanin accumulation by activating the expression of structural genes involved in the anthocyanin biosynthetic pathway [46,47]. However, the role of R2R3-MYB TFs in regulating flower color in *P. suffruticosa* has seldom been functionally verified. Therefore, determining how certain R2R3-MYB TFs regulate anthocyanin production in *P. suffruticosa* would aid in breeding improved cultivars with desirable flower colors.

In the present study, two novel R2R3-MYB TFs (*PsMYB114L* and *PsMYB12L*) possibly involved in anthocyanin biosynthesis were successfully cloned and characterized from the petals of *P. suffruticosa* ‘Shima Nishiki’ and found to contain full-length cDNA of 600 and 1140 bp encoding 199 and 379 amino acids, respectively. The amino acid sequence alignment between *PsMYB114L*/*PsMYB12L* and other known R2R3-MYB TFs involved in anthocyanin regulation indicated that the R2R3 domain distributions of these R2R3-MYB TFs were highly similar, but a bHLH-interaction motif ([D/E]Lx2[R/K]x3Lx6Lx3R) existed in the R3 domain of *PsMYB114L*, whereas *PsMYB12L* did not contain this motif for interaction with bHLH proteins. In *Arabidopsis*, based on a similar function, 125 TFs of R2R3-MYB gene-family members were classified into more than 25 subgroups [48]. Furthermore, many previous studies demonstrated that subgroup 7 [49,50], characterized by both the SG7 ([K/R][R/x][R/K]xGRT[S/x][R/G]xx[M/x]K) and SG7-2 ([W/x][L/x]LS) motifs, specifically regulated flavonol biosynthesis. *PsMYB12L* contained these two motifs (SG7 and SG7-2) at the C terminus of the protein, but *PsMYB114L* lacked these two motifs.

Phylogenetic analysis indicated that *PsMYB12L* and 5 flavonol-regulating R2R3-MYB TFs (*VvMYBF1*, *EsMYBF1*, and *AtMYB11/12/111*) belonging to subgroup 7 [30,51] were clustered together, and *PsMYB114L* and certain R2R3-MYB TFs belonging to subgroup 5 (*AtTT2*, *ZmC1*, *VvMYBPA2*, etc.) [52,53,54,55] had relatively higher homology. Based on the motif analysis of amino acid sequences and phylogenetic analysis, *PsMYB114L* might regulate anthocyanin production by combinatorially interacting with a basic helix-loop-helix (bHLH) factor [25,56,57]. *PsMYB12L* might independently regulate the expression of anthocyanin biosynthesis-related genes without the MBW complex [58].

In addition, we conducted further ectopic transgenic studies by overexpressing *PsMYB114L*/*PsMYB12L* in *Arabidopsis* and apple calli. In contrast to the green-colored leaves of the Col-0 *A. thaliana* ecotype and the white-colored WT apple calli, the leaves of these transgenic *Arabidopsis* plants turned purple-red and the transgenic calli of *PsMYB114L* and *PsMYB12L* were red, which was in agreement with their remarkably higher anthocyanin content. The color and total anthocyanin content analyses of *Arabidopsis* and apple calli indicated that these two R2R3-MYB TFs contribute to anthocyanin accumulation in transgenic lines.

Subsequently, qRT-PCR analysis of seven anthocyanin biosynthesis-related genes (*MdCHS*, *MdCHI*, *MdF3H*, *MdF3′H*, *MdDFR*, *MdANS, MdFLS*, and *MdANR*) was further performed in *Arabidopsis* and apple calli. In terms of *PsMYB114L*, the qRT-PCR results in *Arabidopsis* showed that the expression levels of *AtDFR* and *AtANS* were significantly upregulated, whereas *AtFLS* and *AtANR* were downregulated to a certain extent compared with the levels in the Col-0. Furthermore, the qRT-PCR results in apple calli showed that the expression levels of *MdDFR* and *MdANS* were upregulated to a certain extent, whereas *MdFLS* and *MdANR* (especially *MdFLS*) were significantly downregulated compared with the levels in the WT. Meanwhile, based on the results of expression patterns of *PsMYB114L* in *P. suffruticosa* ‘Shima Nishiki’, we have known that *PsMYB114L* have a positive correlation with *PsFLS* and *PsANR*. By comparing these three qRT-PCR results in *Arabidopsis*, apple calli, and *P. suffruticosa*, we found differences in the expression patterns of some anthocyanin biosynthesis-related genes. Previous studies have showed that many R2R3-MYB TFs usually regulate flavonoid biosynthesis by interacting with the promoter of the targeted structural genes [55,58]. For promoter region, in general, the sequence of the same structural gene in different plant species also differs greatly. Therefore, it is possible that the same MYB TFs performed different regulatory mechanisms of flavonoid biosynthesis in different species [59]. Dihydroflavonol is the direct substrate for two key genes (*FLS* and *DFR*) in the flavonoid biosynthetic pathways, and these two genes usually show a competitive interaction in producing colored anthocyanidin and colorless flavonols [60]. In this study, the strong upregulation of *AtDFR* and *AtANS* may have played key roles in activating the branch of the anthocyanin biosynthesis, resulting in purple-red leaves in the transgenic *Arabidopsis* plants of *PsMYB114L*, whereas the strong downregulation of *MdFLS* would inhibit the branch of the flavonol biosynthesis, resulting in the production of anthocyanins and a red-colored phenotype in the transgenic calli of *PsMYB114L.* Furthermore, because *PsMYB114L* has a bHLH-interaction motif, it may form an MBW complex and contribute to anthocyanin accumulation by regulating the expression of these key genes (*AtDFR*, *AtANS*, *MdFLS*, and *MdANR*) in *Arabidopsis* and apple calli.

With regard to *PsMYB12L*, the qRT-PCR results showed that the expression levels of the four genes (*AtCHS*, *AtCHI*, *AtDFR* and *AtANS*) were all significantly upregulated in the transgenic *Arabidopsis* plants. Moreover, the expression levels of *MdDFR* and *MdANS* were significantly upregulated in the transgenic calli of *PsMYB12L*, but *MdFLS* was significantly downregulated. Meanwhile, based on the results of expression patterns of *PsMYB12L* in *P. suffruticosa* ‘Shima Nishiki’, we have known that *PsMYB12L* have a positive correlation with *PsDFR*, *PsANS*, and *PsF3′H*. By comparing these three qRT-PCR results in *Arabidopsis*, apple calli, and *P. suffruticosa*, we can found that the two key anthocyanin biosynthesis-related genes (*DFR* and *ANS*) showed a very similar expression pattern. We considered that *PsMYB12L* should be a specific transcriptional regulator on *DFR* and *ANS* genes in these three species. Furthermore, we also found differences in the expression patterns of the *FLS* gene in *Arabidopsis* and apple calli, and considered that the expression difference of the *FLS* gene is likely caused by the promoter sequence specificity of this gene in these two species [61]. Based on the motif analysis of *PsMYB12L*, we speculated that the TF may be a flavonol-specific MYB regulator. Many flavonol-specific MYB TFs have been isolated and functionally verified in various plants, such as *A. thaliana*, *Vitis vinifera*, and *Epimedium sagittatum* [50,62,63]. Furthermore, many flavonol-specific MYB TFs negatively regulate anthocyanin accumulation by inducing higher expression of the *FLS* gene. By overexpressing *AtMYB12* in tobacco, the expression of *NtCHS*, *NtCHI*, and *NtFLS* was specifically activated; moreover, the flowers of the transgenic plants were paler in color than their wild-type counterparts [64]. Ectopic expression analysis of *EsMYBF1* in transgenic tobacco indicated that *NtCHS*, *NtCHI*, *NtF3H*, and *NtFLS* were upregulated but *NtDFR* and *NtANS* were significantly downregulated, and the accumulation of anthocyanins in transgenic tobacco flowers was also remarkably decreased [63]. A study on the overexpression of *PpMYB15* in tobacco showed that it can significantly activate the expression of *NtCHS*, *NtCHI*, *NtF3H*, and *NtFLS*, while it had no effects on the expression of *NtDFR* and *NtANS*, resulting in pale-pink or pure white flowers in transgenic tobacco plants [40]. Compared with the expression of anthocyanin biosynthesis-related genes documented in the abovementioned studies, in this study *AtCHS/MdCHS*, *AtCHI*, *AtF3H/MdF3H*, and *AtFLS* had a somewhat similar expression pattern and *MdFLS*, *AtDFR/MdDFR* and *AtANS/MdANS* exhibited the opposite pattern. However, the lower expression of the *MdFLS* gene and the higher expression of *AtDFR/MdDFR* and *AtANS/MdANS* were consistent with the significantly higher anthocyanin accumulation in transgenic lines of *PsMYB12L*. Beacuse *PsMYB12L* has the flavonol-specific motif and lacks the bHLH-interaction motif, it alone enhances anthocyanin production by regulating the expression of these key genes (*AtDFR*/*MdDFR*, *AtANS/MdANS*, and *MdFLS*) independently of bHLH cofactors in *Arabidopsis* and apple calli.

## 4. Materials and Methods

### 4.1. Plant Materials

The tree peony cultivar *P. suffruticosa* ‘Shima Nishiki’ was grown in the experimental nursery of Forestry College, Shandong Agricultural University, Tai’an, Shandong, China. Flower samples were collected at five early flower-bud developmental stages (flower bud emerging stage (S1), small bell-like flower-bud stage (S2), large bell-like flower-bud stage (S3), bell-like flower-bud extending stage (S4), and color exposing stage (S5)) (Appendix A) [65]. All these samples were immediately frozen in liquid nitrogen and then stored at –80 °C for further experiments.

The *A. thaliana* ecotype Columbia (Col-0) was used for genetic transformation and phenotypic analysis in the present study. The plants were grown under a 16 h light/ 8 h dark photoperiod at 23 °C/21 °C

Furthermore, calli of the wild type (WT) of ‘Orin’ apple were subcultured on Murashige and Skoog (MS) medium with 1.5 mg L^−1^ 6-benzyl adenine (6-BA) and 0.5 mg L^−1^ 2,4-dichlorophenoxyacetic acid (2,4-D) at room temperature (24 °C) in a continuous dark environment at 15-day intervals [66]. Subsequently, the calli were used for genetic transformation and phenotypic analysis.

### 4.2. Total RNA Extraction and cDNA Synthesis

Total RNA was extracted from all samples according to instructions of the EASY Spin Plant RNA Rapid Extraction Kit (Aidlab Biotech, Beijing, China). The purity and concentration of all RNA samples were assessed using a Nanodrop 2000C spectrophotometer (Thermo Fisher Scientific, Wilmington, Delaware, DE, USA), and RNA quality was detected using 1 % agarose gels. Furthermore, cDNA was synthesized with 1 µg of total RNA using 5× All-In-One RT MasterMix (with an AccuRT Genomic DNA Removal Kit) (ABM, Vancouver, BC, Canada).

### 4.3. Cloning of the PsMYB114L and PsMYB12L Genes in P. suffruticosa

In this study, based on the transcriptome sequencing data of *P. suffruticosa* ‘Shima Nishiki’ in our laboratory, two R2R3-MYB transcription factors were filtered by analyzing the functional annotations of MYB unigenes and performing gene expression analysis.

The cDNA of the ‘Shima Nishiki’ cultivar’s petals was used as the template. The full-length coding sequence (CDS) of the *PsMYB114L* (MK518073) and *PsMYB12L* (MK518074) genes was amplified using PCR. The complete 5′ CDS of the *PsMYB114L* and *PsMYB12L* genes was identified from the transcriptome sequencing data of *P*. *suffruticosa* ‘Shima Nishiki’. The cDNA 3′ end sequence of these candidate genes was obtained using nested PCR technology using *PsMYB114L*-1-F/*PsMYB114L*-2-F and *PsMYB12L*-1-F/*PsMYB12L*-2-F as forward primers (Appendix A), respectively, and B26 was used as the common reverse primer. The full-length cDNA of the *PsMYB114L* and *PsMYB12L* genes was amplified with the forward primers *PsMYB114L*-F1/*PsMYB12L*-F1 and the reverse primers *PsMYB114L*-R1/*PsMYB12L*-R1 (Appendix A). The PCR program of gene amplification was as follows: initial denaturation at 95 °C for 1 min, followed by 30 cycles of 98 °C for 10 s, 60 °C for 15 s and 68 °C for 60 s. The PCR products were purified and cloned into the pTOPO-Blunt Simple vector for sequencing.

### 4.4. Subcellular Localization

The full-length cDNA without the termination codon of *PsMYB114L*/*PsMYB12L* was amplified with special primers (*PsMYB114L*-GFPF/*PsMYB12L*-GFPF and *PsMYB114L*-GFPR/*PsMYB12L*-GFPR) (Appendix A) with restriction sites (*Xba* I and *Kpn* I) and subcloned into the pCAMBIA1301-GFP vector between the *Xba* I and *Kpn* I sites to create the *PsMYB114L*-GFP/*PsMYB12L*-GFP fusion construct. The recombinant vectors (*PsMYB114L*-GFP/*PsMYB12L*-GFP) and control vector (pCAMBIA1301-GFP) were then introduced into tobacco leaves by agroinfiltration. These infiltrated plants were grown for over 72 h in a growth chamber, and the GFP fluorescence of samples was observed under a Nikon C2-ER confocal laser scanning microscope (Nikon, Tokyo, Japan) [67].

### 4.5. Overexpression Vector Construction

The full-length cDNA of the *PsMYB114L* and *PsMYB12L* genes from the petals of *P. suffruticosa* ‘Shima Nishiki’ was amplified using recombinant primers (*PsMYB114L*-F2/*PsMYB12L*-F2 and *PsMYB114L*-R2/*PsMYB12L*-R2) (Appendix A) with restriction sites (*Spe* I and *BstE* II). Based on the predesigned vector construction procedure, the pCAMBIA1304 empty vector and the pTOPO-Blunt Simple vector containing the target genes (*PsMYB114L* and *PsMYB12L*) with restriction sites were double digested separately between the *Spe* I and *BstE* II sites and then recombined (Appendix A). Subsequently, the two recombinant vectors pCAMBIA1304-*PsMYB114L* (Appendix A) and pCAMBIA1304-*PsMYB12L* (Appendix A) were verified successfully by PCR and sequencing with the forward vector validation primer 1304Ve-F and the reverse primers *PsMYB114L*-R2/*PsMYB12L*-R2 (Appendix A). These two overexpression constructs were also introduced into *Agrobacterium tumefaciens* strain GV3101 using the freeze-thaw method.

### 4.6. Stable Transformation of Arabidopsis

The transformation of *Arabidopsis* was performed using the floral dip transformation method [68]. An *A*. *tumefaciens* infection solution (OD600 = 0.8–1.2) containing 5 % sucrose and 0.01 % Silwet L-77 was prepared to infect inflorescences, and the infection time per inflorescence was 15 s. Subsequently, these plants were transferred to a dark treatment for 24 h. These steps were repeated twice more according to the growth state of the plant. Mature T1 seeds were harvested, surface sterilized, and then sown on MS medium with 30 mg L^−1^ hygromycin B to screen for positive transformants. The resistant seedlings were transplanted into soil and then placed in a light incubator (16 h light/8 h dark at 23 °C/21 °C). When these transgenic *Arabidopsis* plants had grown to a certain size, they were further verified with gene-specific primers by PCR.

### 4.7. Stable Transformation of Apple Calli

To transform apple calli, 15-day-old WT apple calli were incubated with *A*. *tumefaciens* infection solution that carried pCAMBIA1304-PsMYB114L/pCAMBIA1304-PsMYB12L for 20 min, and the apple calli were then cocultured on MS medium supplemented with 0.5 mg L^−1^ 2,4-D and 1.5 mg L^−1^ 6-BA for 2 days at 24 °C in the dark. Subsequently, the apple calli were washed three times with sterile water and transferred to a selective medium that contained 15 mg L^−1^ hygromycin B for transgene selection. The transgenic apple calli were cocultured in the selective medium containing appropriate concentrations of an antibiotic and transferred to a light incubator with constant light (photon flux density of ~100 μmol s^−1^ m^−2^) and low-temperature (15 °C) treatments for phenotypic observation [69,70].

### 4.8. Measurement of Total Anthocyanin Content

Total anthocyanin were extracted from the rosette leaves of 25-day-old *Arabidopsis* plants and apple calli cultured for 7 days. Anthocyanin extraction was performed using a methanol–HCl method [71]. Approximately 0.1 g of each sample was incubated in 5 mL of 0.1 % acidic methanol solution (CH_3_OH:HCl:H_2_O = 70:0.1:29.9, *v*/*v*/*v*) overnight in the dark at 4 °C. The absorbance of each extract was measured at 530 and 657 nm with a UV-1600 spectrophotometer (SHIMADZU, Kyoto, Japan). The total anthocyanin content was calculated using the following equation: Q_Total Anthocyanin_ = (A530 − 0.25 × A657) × FM^−1^. There were three biological replicates for each sample.

### 4.9. Quantitative Real-Time PCR (qRT-PCR) Analysis

qRT-PCR was performed to analyze the expression levels of anthocyanin biosynthesis-related genes in all plant materials in this study. The qRT-PCR experiments were conducted using SYBR^®^ Premix Ex Taq™ (Tli RNaseH Plus) (TaKaRa, Kyoto, Japan) on a Bio-Rad CFX96™ Real-Time system (Bio-Rad, Hercules, CA, USA) with three biological replicates according to the manufacturer’s instructions. The PCR conditions were as follows: 95 °C for 30 s, 40 cycles of 95 °C for 5 s and 60 °C for 30 s and then a dissociation stage at 95 °C for 10 s, 65 °C for 5 s and 95 °C for 5 s. The *Psubiquitin* gene, *AtActin2* gene and *MdActin* gene were used as internal controls to normalize the expression levels in *P*. *suffruticosa*, *A. thaliana* and *Malus domestica*, respectively. All gene-specific primers used in this study are shown in Appendix A [39,66]. The relative expression levels of genes were calculated using the 2^−ΔΔ*C*t^ method [72].

### 4.10. Sequence and Statistical Analysis

Multiple sequence alignment was performed using DNAMAN 8.0 software (Lynnon Biosoft, San Ramon, CA, USA). Homology search of sequences was carried out using the GenBank BLAST. Phylogenetic tree construction of sequences was performed using MEGA 5.0 software (Arizona State University, Tempe, AZ, USA) with the bootstrap values from 1000 replicates. Primers were designed using Primer Premier 5.0 software (PREMIER Biosoft International, Palo Alto, CA, USA). All experiments were repeated three times, and the data are expressed as the mean ± standard error. Variance analyses were performed using SPSS software ver. 17.0 (SPSS Inc., Chicago, IL, USA). *p*-values of < 0.05 were considered statistically significant.

## 5. Conclusions

In conclusion, two novel R2R3-MYB TFs, namely *PsMYB114L* and *PsMYB12L*, were successfully cloned from the petals of *P. suffruticosa* ‘Shima Nishiki’ and functionally characterized by heterologous expression in *Arabidopsis* and apple calli. Based on the above results, we preliminarily demonstrated the potential functional roles of *PsMYB114L* and *PsMYB12L* in regulating anthocyanin biosynthesis. These results provide a valuable resource for further understanding the molecular regulatory mechanisms of anthocyanin biosynthesis and accumulation in *P. suffruticosa* and breeding improved cultivars of *P. suffruticosa* with desirable flower colors in the future.

## Figures and Tables

**Figure 1 ijms-20-01055-f001:**
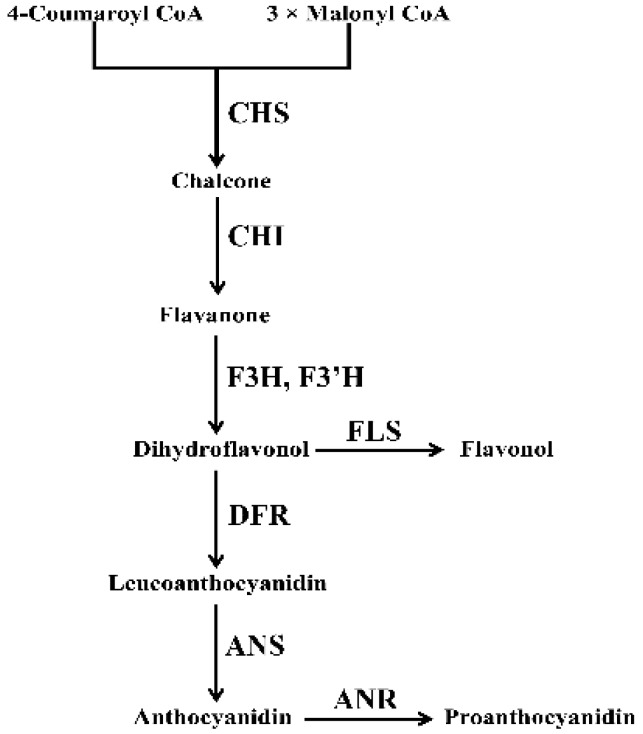
A general schematic diagram of the metabolic pathway related to anthocyanin biosynthesis. CHS, chalcone synthase; CHI, chalcone isomerase; F3H, flavanone 3-hydroxylase; F3′H, flavonoid 3′-hydroxylase; DFR, dihydroflavonol 4-reductase; ANS, anthocyanidin synthase; FLS, flavonol synthase; ANR, anthocyanidin reductase.

**Figure 2 ijms-20-01055-f002:**
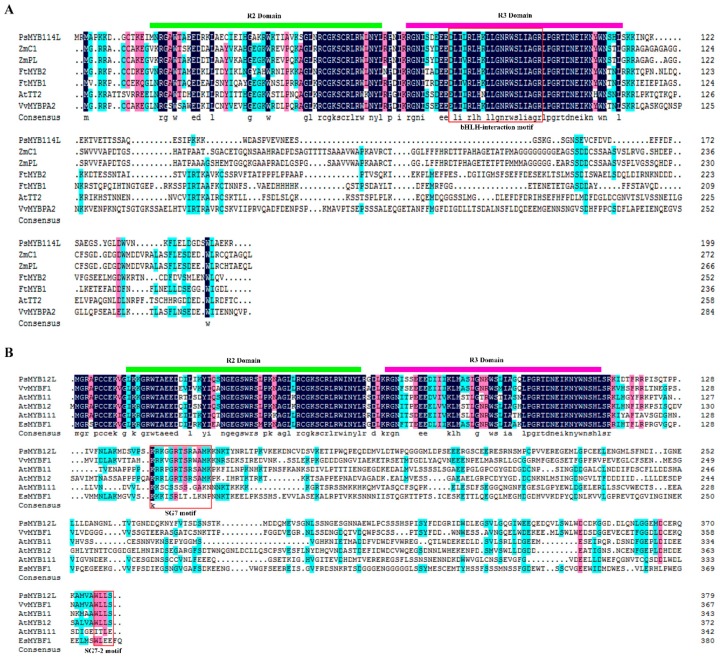
Amino acid sequence alignment analysis of the *PsMYB114L* (**A**) and *PsMYB12L* (**B**) genes with other known R2R3-MYB TFs. The green and pink long lines indicate the R2 and R3 domain, respectively. The red boxes show the conserved bHLH-interaction, SG7 and SG7-2 motifs. The NCBI GenBank accession numbers of these sequences are as follows: *ZmC1*, *Zea mays*, AF320613.3; *ZmPL*, *Zea mays*, NM_001112415.1; *FtMYB2*, *Fagopyrum tataricum*, JF313346.1; *FtMYB1, Fagopyrum tataricum*, JF313344.1; *AtTT2*, *Arabidopsis thaliana*, NM_122946.3; *VvMYBPA2*, *Vitis vinifera*, NM_001281024.1; *VvMYBF1*, *Vitis vinifera*, FJ948477.2; *AtMYB11*, *Arabidopsis thaliana*, NM_116126.3; *AtMYB12*, *Arabidopsis thaliana*, NM_130314.4; *AtMYB111*, *Arabidopsis thaliana*, NM_124310.3; *EsMYBF1*, *Epimedium sagittatum*, KU365320.1.

**Figure 3 ijms-20-01055-f003:**
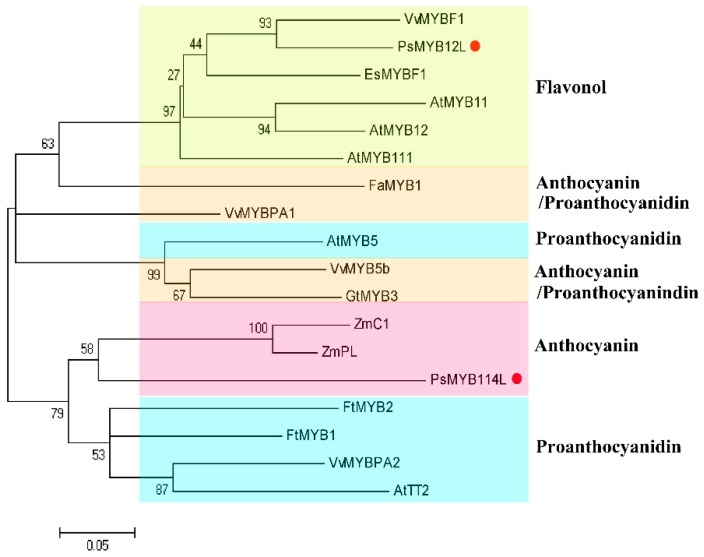
Phylogenetic analysis of the *PsMYB114L* and *PsMYB12L* genes with R2R3-MYB TFs from other species. The NCBI GenBank accession numbers of these sequences are as follows: *FaMYB1*, *Fragaria x ananassa*, AF401220.1; *VvMYBPA1, Vitis vinifera*, NM_001281231.1; *VvMYB5b*, *Vitis vinifera*, NM_001280925.1; *AtMYB5*, *Arabidopsis thaliana*, AF401220.1; *GtMYB3*, *Gentiana triflora*, AB289445.1; *ZmC1*, *Zea mays*, AF320613.3; *ZmPL*, *Zea mays*, NM_001112415.1; *FtMYB2*, *Fagopyrum tataricum*, JF313346.1; *FtMYB1, Fagopyrum tataricum*, JF313344.1; *AtTT2*, *Arabidopsis thaliana*, NM_122946.3; *VvMYBPA2*, *Vitis vinifera*, NM_001281024.1; *VvMYBF1*, *Vitis vinifera*, FJ948477.2; *AtMYB11*, *Arabidopsis thaliana*, NM_116126.3; *AtMYB12*, *Arabidopsis thaliana*, NM_130314.4; *AtMYB111*, *Arabidopsis thaliana*, NM_124310.3; *EsMYBF1*, *Epimedium sagittatum*, KU365320.1.

**Figure 4 ijms-20-01055-f004:**
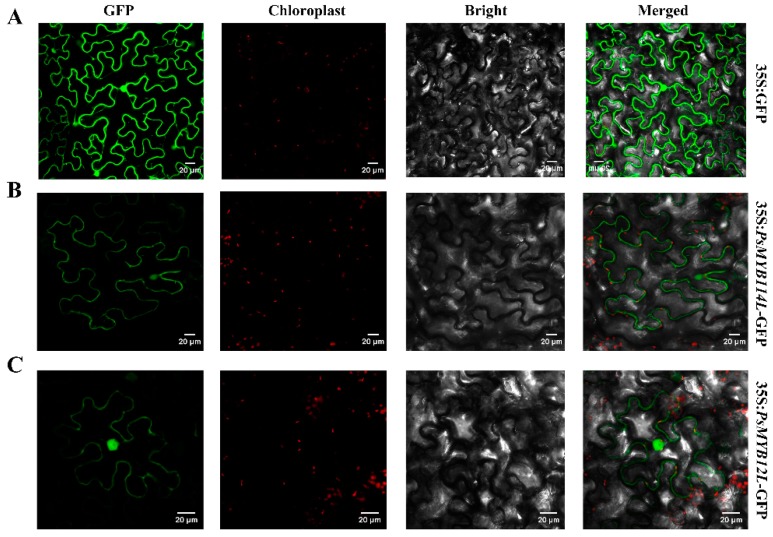
Subcellular localization analysis of the *PsMYB114L* and *PsMYB12L* genes. (**A**) Control vector (pCAMBIA1301-GFP) expressed in epidermal cells of tobacco leaves. (**B**) Recombinant vector (*PsMYB114L*-GFP) expressed in epidermal cells of tobacco leaves. (**C**) Recombinant vector (*PsMYB12L*-GFP) expressed in epidermal cells of tobacco leaves. White lines at the bottom right of the picture represent 20 μm in the respective pixel. GFP, GFP fluorescence; Chloroplast, Chloroplast fluorescence; Bright, Bright field; Merged, Superposition of bright field and fluorescence.

**Figure 5 ijms-20-01055-f005:**
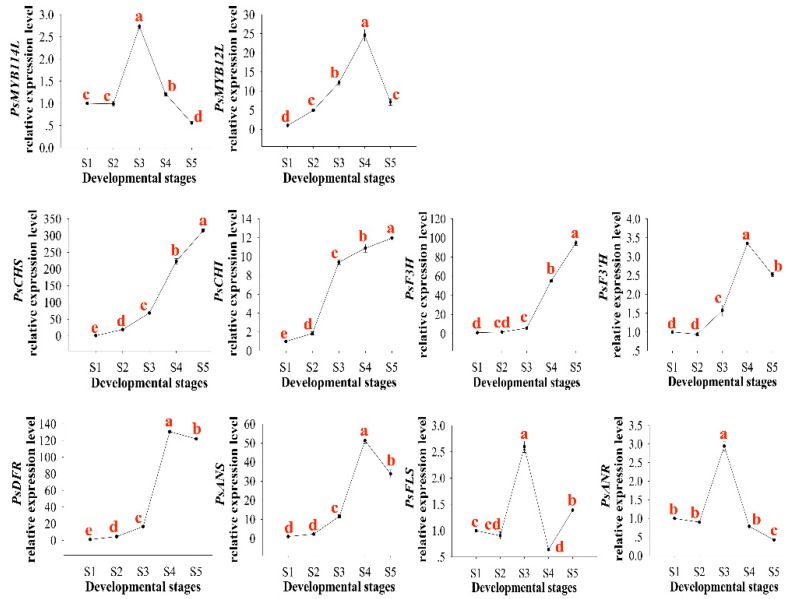
The expression patterns of the *PsMYB114L* gene, *PsMYB12L* gene and anthocyanin biosynthesis-related structural genes in *P. suffruticosa* ‘Shima Nishiki’. S1, flower bud emerging stage; S2 small bell-like flower-bud stage; S3, large bell-like flower-bud stage; S4, bell-like flower-bud extending stage; S5, color exposing stage. Different lowercase letters indicate significant differences at *p* < 0.05.

**Figure 6 ijms-20-01055-f006:**
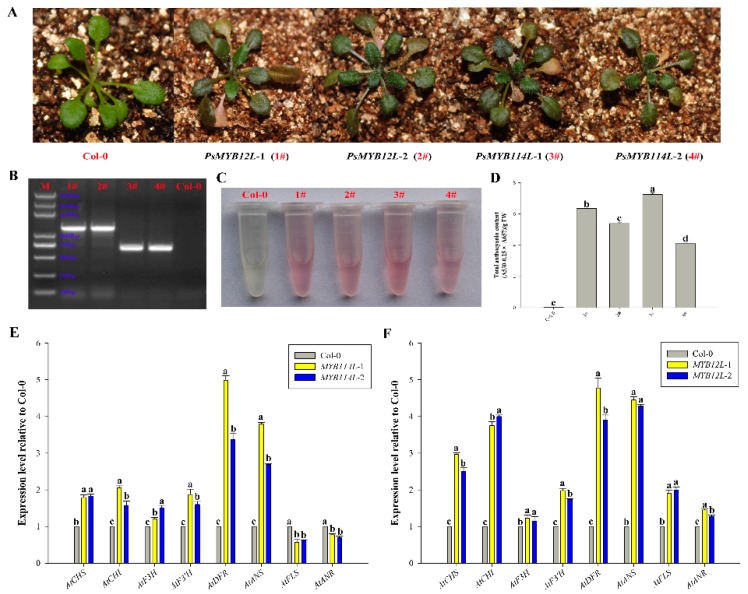
Overexpression analysis of the *PsMYB114L* and *PsMYB12L* genes in *Arabidopsis*. (**A**) Comparison of leaf colors in transgenic *Arabidopsis* plants and Col-0. (**B**) Results of positive PCR detection in transgenic *Arabidopsis* plants. (**C**) Anthocyanin extraction solutions for transgenic *Arabidopsis* plants and Col-0. (**D**) Total anthocyanin content in transgenic *Arabidopsis* plants and Col-0. (**E**) Expression analysis of anthocyanin biosynthesis-related genes in transgenic *Arabidopsis* plants of *PsMYB114L* and Col-0. (**F**) Expression analysis of anthocyanin biosynthesis-related genes in transgenic *Arabidopsis* plants of *PsMYB12L* and Col-0. Col-0, *Arabidopsis thaliana* ecotype Columbia; 1# and 2#, two transgenic lines of the *PsMYB12L* gene; 3# and 4#, two transgenic lines of the *PsMYB114L* gene. Different lowercase letters indicate significant differences at *p* < 0.05.

**Figure 7 ijms-20-01055-f007:**
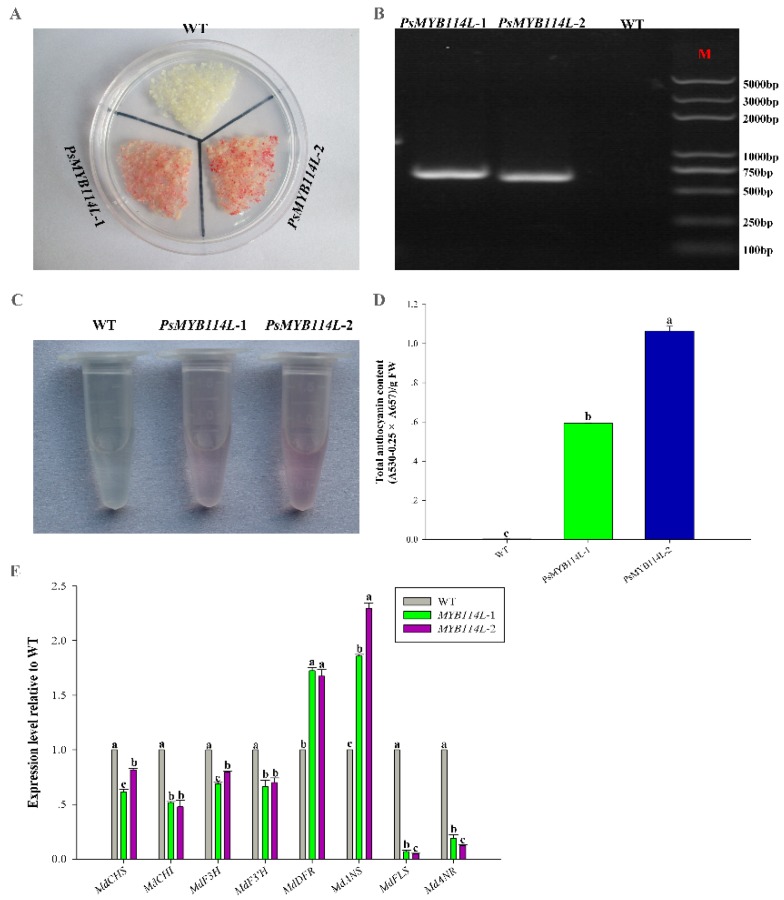
Overexpression analysis of the *PsMYB114L* gene in apple calli. (**A**) Colors observed in transgenic apple calli and the WT. (**B**) Results of positive PCR detection in transgenic apple calli. (**C**) Anthocyanin extraction solutions for transgenic apple calli and the WT. (**D**) Total anthocyanin content in transgenic apple calli and the WT. (**E**) Expression analysis of anthocyanin biosynthesis-related genes in transgenic apple calli and the WT. WT, Wild-type ‘Orin’ apple calli; *PsMYB114L*-1 and *PsMYB114L*-2, two transgenic lines of the *PsMYB114L* gene. Different lowercase letters indicate significant differences at *p* < 0.05.

**Figure 8 ijms-20-01055-f008:**
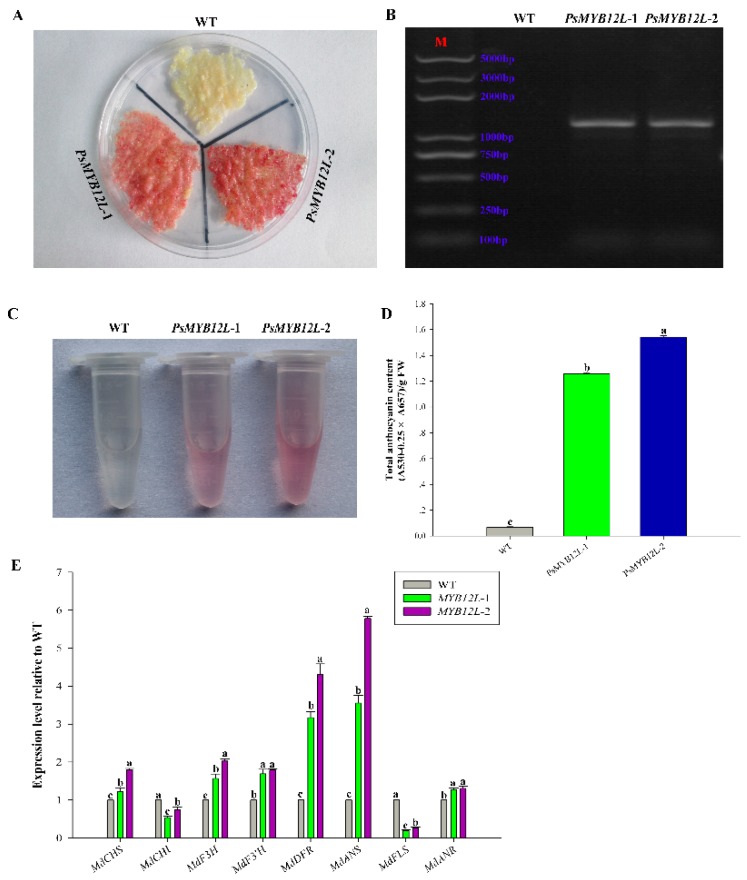
Overexpression analysis of the *PsMYB12L* gene in apple calli. (**A**) Colors observed in transgenic apple calli and the WT. (**B**) Results of positive PCR detection in transgenic apple calli. (**C**) Anthocyanin extraction solutions for transgenic apple calli and the WT. (**D**) Total anthocyanin content in transgenic apple calli and the WT. (**E**) Expression analysis of anthocyanin biosynthesis-related genes in transgenic apple calli and the WT. *PsMYB12L*-1 and *PsMYB12L*-2, two transgenic lines of the *PsMYB12L* gene. Different lowercase letters indicate significant differences at *p* < 0.05.

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
