# Peer review of "Identification of Two Novel R2R3-MYB Transcription factors, PsMYB114L and PsMYB12L, Related to Anthocyanin Biosynthesis in Paeonia suffruticosa"

_ijms, 2019, doi:10.3390/ijms20051055_

Round 1

Reviewer 1 Report

General comments:

The manuscript is very well written. The function of the identified R2-R3 Myb TF from Paeonia suffruticosa were extensively proved by overexpression in Arabidopsis and Apple calli. qPCR of anthocyanin biosynthesis related genes indicate interaction with key genes of anthocyanin accumulation as well as interaction with key genes of the flavonoid biosynthetic pathway.

Comments:

Abstract

L22 – please make clearer, that transgenesis of Arabidopisis and appli calli forms part of this manuscript (otherwise it sounds as if authors were citing another work)

Introduction

L52 - These regulatory genes - better: the regulatory genes

Two colored P. suffruticosa ‘Shima Nishiki’ was used in this research – it would be interesting to include information about knowledge on the origin of colour differences (Zang et al 2018 3 Biotech. 2018 Oct;8(10):420)

Material and Methods

L. 275-280 – did authors use petals with both pink and red colour -please indicate (see double colour variety)

L 324, L333 - A. tumefaciens – italics

L342 – please indicate how many samples were used for anthocyanin content analysis and which leaves (age, position) were used.

L351 – does that confer to biological or technical replicas – please indicate

Author Response

Dear reviewer,

Thank you very much for your comments on our manuscript entitled “Identification of two novel R2R3-MYB transcription factors, PsMYB114L and PsMYB12L, related to anthocyanin biosynthesis in Paeonia suffruticosa” (ijms-446453). We appreciate the comments from you and have made sincere efforts to address them in the revised manuscript. The manuscript was carefully revised according to the comments and suggestions, and our point-by-point responses to your comments are listed below. The line numbers refer to those in the revised manuscript without track changes. Furthermore, we have used a differently colored font for all changes in the revised manuscript with track changes (blue words or sentences with strikeout indicate that the words or sentences were deleted, and red words or sentences indicate that the words or sentences were changed or added).

1.Abstract: L22 – please make clearer, that transgenesis of Arabidopisis and appli calli forms part of this manuscript (otherwise it sounds as if authors were citing another work).

Response: Thank you very much for your suggestion. We have clarified the description of the transgenesis of Arabidopisis and apple calli in the Abstract section of revised manuscript (Lines 26-31).

2. Introduction: L52 - These regulatory genes - better: the regulatory genes.

Response: Thank you very much for your suggestion. We have modified “These regulatory genes” to “the regulatory genes” in the revised manuscript (Line 72).

3.Introduction: Two colored P. suffruticosa ‘Shima Nishiki’ was used in this research – it would be interesting to include information about knowledge on the origin of colour differences (Zhang et al 2018 3 Biotech. 2018 Oct;8(10):420).

Response: Considering the reviewer’s suggestion, we have added some basic information about knowledge on the origin of colour differences of P. suffruticosa ‘Shima Nishiki’ to the revised manuscript (Lines 54-57).

4. Material and Methods: L. 275-280 – did authors use petals with both pink and red colour -please indicate (see double colour variety).

Response: In this study, the petal samples were collected at five early flower bud stages before color exposing. During these flower-bud developmental stages, the petals’ red and pink colors cannot be distinguished yet. Therefore, we sampled the whole petals without considering this aspect of petal colors in this study.

5.Material and Methods: L 324, L333 - A. tumefaciens – italics

Response: We apologize for this inaccuracy and have modified the term “A. tumefaciens” to “A. tumefaciens” in the revised manuscript (Line 409).

6.Material and Methods: L342 – please indicate how many samples were used for anthocyanin content analysis and which leaves (age, position) were used.

Response: We apologize that this was not clear in the original manuscript. For Arabidopsis/apple calli, there were three biological replicates used for anthocyanin content analysis,. The samples were collected from the rosette leaves of 25-day-old Arabidopsis plants and apple calli cultured for 7 d. The details of the samples have been described in the revised manuscript (Lines 429-430 and 436).

7.Material and Methods: L351 – does that confer to biological or technical replicas – please indicate

Response: We apologize that this information was not clear in the original manuscript, and this refers to biological replicates. It has been described in the revised manuscript (Line 441).

Thank you and best regards.

Reviewer 2 Report

In the manuscripr “Identification of two novel R2R3-MYB transcription factors, PsMYB114L and PsMYB12L, related to anthocyanin biosynthesis in Paeonia suffruticosa” the authors identified, cloned and functionally analysed two R2R2 MYB transcription factors playing a role in anthocyanin biosynthesis. The work provides new insights into two genes possibly involved in petal colours in an economically important species. The two transcription factors studied cause an increase in anthocyanin production when overexpressed in heterologous systems, however the attempts to understand the mechanisms remains in my opinion quite vague and confusing.

In general, in every results section, the authors should use more words to describe why they performed the experiments (the choices they made) and possibly, at the end of each section, what the take home messages are.

English should be improved. I will point out some examples along the manuscript but I suggest a careful a thorough editing work.

Abstract:

Line14-15: I find it repetitive (ornamental is used twice). I would rephrase as follow: Flower color is a charming phenotype with very important ornamental and commercial values.

Line 15-16: : I find it repetitive (anthocyanin is used twice). Anthocyanins play critical roles in determining flower color pattern formation and their biosynthesis is normally regulated…..

Line 21: substitute “and” with “whereas” to emphasise the differences between the two TFs

Line 23: “and of PSMYB12L”

Line 24: “phenotypes” is vague. Unclear in the abstract that we are looking at Arabidopsis leaf colour for example

Introduction

Line 41: what is intended with “produced at home”? private homes or China? Rephrase as e.g. About 1500 cultivars of P. suffruticosa with a variety of flower colours have been produced by breeders worldwide.

Line 54 and line 57 replace “these” (structural/regulatory genes) with “the”.

Overall more and clearer background about what is know for the anthocyanin biosynthetic pathway shoud be given. E.g. Line 143: this is the first time genes like FLS and ANR are mentioned for the first time in line 143 (results) without any explanations.

Results

2.1 It is unclear how the genes were selected, the primers designed and genes amplified (the details should go in M&M but in the result section I would like to read the rationale behind the choices). In materials and methods, the authors said they have transcriptome data? Were then these 2 genes directly chosen from the sequencing data? If so, why (were DEG genes in the transcriptomic study?)? How were the names given? I saw that the same authors have already published the transcriptomic work on which I believe this further characterization presented in this manuscript is based upon. The work is cited in ref 50. I believe more reference to that study as a starting point would make it clearer to the readers.

Figure 1 could be something for supplementary and the sequences deposited in public databases (e.g. ncbi) if not already public give reference 50.

In the amino acid sequence alignment, it should be explained what the SAINT superfamily is and what the conservation of the domains means. The bHLH-interaction motif is clear, but what are SG7 and SG7-2? What is their biological function, if known, where is a reference to this knowledge?

How were the genes for the phylogenetic analysis chosen? What is the interpretation of the results? Is it known what the role of the genes included in the analysis is?

2.2 Subcellular localisation. Also here a conclusion/comments would help. I would expect TFs to be nuclear localised. Is the localization found surprising? How does this relate to other localization of R2R3 MYB TFs?  Is it possible to include negative controls?

2.3 Expression analyses

Line 140: “collected at 5 developmental stages” (à refer to  M&M and figure S2)

In figure 5 put the expression of PsMYB114L and PsMYB12L above and then the graphs of the other genes below (as described in the text). Increase font size in the graphs as it is hardly possible to read.

Line 143: this is the first time genes like FLS and ANR are mentioned (a part from the abstract). Their known role should be explained in the introduction.

2.4 overexpression in Arabidopsis.

Expression of candidate genes in transgenics Arabidopsis should be performed and compared with the expression analyses done in Apple calli to have stronger indications of the reliability of the results obtained in the latter.

The CDS of the 2 TFs were isolated in Paeonia (which tissue?). Their expression has been tested in different stages of the petals. What is the phenotype of Arabidopsis flowers? Do they accumulate anthocyanins and do they present a coloured phenotype? 

Line 152 to 158 belongs to material and methods section.

2.5 overexpression in Apple calli. explain the choice of apple calli system (could Paeonia´s calli be an option?). Explain why “light and low temperature treatments were performed” (plus explain in the M&M the details of these treatments)

Discussion

Integrate and elaborate the results from 2.3 with the results from 2.4 (and of the Arabidopsis qPCR I requested above). e.g. in 2.3 114L peaks in stage 3 as FLS and ANR suggesting perhaps a positive correlation between the genes. However in 2.4 upon overexpression of 114L, FLS and ANR are strongly downregulated. The authors should try to interpret these and other results.

Does it make sense that most structural genes for anthocyanin biosynthesis are downregulated in 35S:114L but still these calli produce more anthcyanins? Authors should comment and interpret these results, surely also by adding more background information of the genes analysed.

Line 218: change “and” with “whereas” or “while” to show contrasts. bHLH-interaction motif ([D/E]Lx2[R/K]x3Lx6Lx3R) existed in the R3 domain of PsMYB114L, whereas PsMYB12L did not contain this motif for interaction with bHLH proteins.

Line 233: remove “,respectively” otherwise it seems that 114L was overexpressed in Arabidopsis (only) and 12L in apple calli (only)

Materials and Methods

It is unclear how the genes were selected, the primers designed and genes amplified. In materials and methods they said they have the genome? and how the names were given

From which tissue and conditions was obtained the cDNA from which the 2 TFs were amplified and cloned

The description of the phylogenetic analyses is missing

Conclusion

The first two sentences say exactly the same concept, remove one of the two.

I would then add some emphasis on the fact that in P. suffruticosa the two genes are detected in the petals (this manuscript-which might indicate a role in flower color) and, if more information is known from the published transcriptomic study (ref 50), mentioned it here to strengthen this hypothesis.

Author Response

Dear reviewer,

Thank you very much for your comments on our manuscript entitled “Identification of two novel R2R3-MYB transcription factors, PsMYB114L and PsMYB12L, related to anthocyanin biosynthesis in Paeonia suffruticosa” (ijms-446453). We appreciate the comments from you and have made sincere efforts to address them in the revised manuscript. The manuscript was carefully revised according to the comments and suggestions, and our point-by-point responses to your comments are listed below. The line numbers refer to those in the revised manuscript without track changes. Furthermore, we have used a differently colored font for all changes in the revised manuscript with track changes (blue words or sentences with strikeout indicate that the words or sentences were deleted, and red words or sentences indicate that the words or sentences were changed or added).

We sincerely hope that the revised manuscript meets the publication standards for International Journal of Molecular Sciences.

1.Abstract: Line14-15: I find it repetitive (ornamental is used twice). I would rephrase as follow: Flower color is a charming phenotype with very important ornamental and commercial values.

Response: Thank you very much for your suggestion. We have modified this phrasing “Flower color is a very important ornamental trait, and a charming phenotype has great ornamental and commercial value” to “Flower color is phenotypic characteristic with considerable ornamental and commercial value” in the revised manuscript (Line 17).

2.Abstract: Line 15-16: I find it repetitive (anthocyanin is used twice). Anthocyanins play critical roles in determining flower color pattern formation and their biosynthesis is normally regulated…..

Response: Thank you very much for your suggestion. We have modified the word “anthocyanins” to “their” in the revised manuscript (Line 19).

3.Abstract: Line 21: substitute “and” with “whereas” to emphasise the differences between the two TFs

Response: Thank you very much for your suggestion.We have substituted the word “and” with “whereas” in the revised manuscript (Line 25).

4.Abstract: Line 23: “and of PsMYB12L”

Response: Thank you very much for your suggestion. We have added the word “of” between the word “and” and “PsMYB12L” in the revised manuscript (Line 28).

5.Abstract:Line 24: “phenotypes” is vague. Unclear in the abstract that we are looking at Arabidopsis leaf colour for example

Response: We apologize that this was not clear in the original manuscript. We have modified the relevant words and sentences. These details have been described in the revised manuscript (Lines 29-31).

6.Introduction: Line 41: what is intended with “produced at home”? private homes or China? Rephrase as e.g. About 1500 cultivars of P. suffruticosa with a variety of flower colours have been produced by breeders worldwide.

Response: We apologize that this was not clear in the original manuscript. Here, it refers to China, not private homes. To avoide ambiguity in the use of “at home”, we have modified the phrase “at home and abroad” to the word “worldwide” in the revised manuscript (Line 51).

7.Introduction: Line 54 and line 57 replace “these” (structural/regulatory genes) with “the”.

Response: Thank you very much for your suggestion. We have replaced “these” with “the” in the revised manuscript (Lines 66 and 72).

8.Introduction: Overall more and clearer background about what is know for the anthocyanin biosynthetic pathway shoud be given. E.g. Line 143: this is the first time genes like FLS and ANR are mentioned for the first time in line 143 (results) without any explanations.

Response: Thank you very much for your suggestion. For FLS and ANR genes mentioned for the first time, we have added explanations in the Introduction section of revised manuscript (Lines 69-72).

Additionally, we have added a general schematic diagram of the metabolic pathway related to anthocyanin biosynthesis (Figure 1). The details of Figure 1 have been described in the revised manuscript (Lines 76-79).

9.Results: 2.1: It is unclear how the genes were selected, the primers designed and genes amplified (the details should go in M&M but in the result section I would like to read the rationale behind the choices). In materials and methods, the authors said they have transcriptome data? Were then these 2 genes directly chosen from the sequencing data? If so, why (were DEG genes in the transcriptomic study?)? How were the names given? I saw that the same authors have already published the transcriptomic work on which I believe this further characterization presented in this manuscript is based upon. The work is cited in ref 50. I believe more reference to that study as a starting point would make it clearer to the readers.

Response: We apologize that this information was not clear in the original manuscript. Considering the reviewer’s suggestion, we have added the related descriptions of these aspects. The details have been described in the revised manuscript (Lines 98-101; Lines 103-106; Lines 369-371; Lines 381-382; Lines 450 and 452).

10.Results: 2.1: Figure 1 could be something for supplementary and the sequences deposited in public databases (e.g. ncbi) if not already public give reference 50.

Response: Considering the reviewer’s suggestion, we have moved Figure 1 to the Supplementary Materials, and renamed it Figure S1. Furthermore, the sequences of the PsMYB114L and PsMYB12L genes have been deposited in public databases of NCBI and their accession numbers are MK518073 and MK518074, respectively. Their accession numbers have been provided in the revised manuscript (Line 106).

11.Results: 2.1: In the amino acid sequence alignment, it should be explained what the SAINT superfamily is and what the conservation of the domains means. The bHLH-interaction motif is clear, but what are SG7 and SG7-2? What is their biological function, if known, where is a reference to this knowledge?

Response: Thank you very much for your suggestion. We have explained what the SAINT superfamily is, what the conservation of the domains means and what their biological functions of SG7 and SG7-2 motifs are in the revised manuscript (Lines 112-114; Lines 115-117; Line 121).

12.Results: 2.1: How were the genes for the phylogenetic analysis chosen? What is the interpretation of the results? Is it known what the role of the genes included in the analysis is?

Response: Based on the relevant conferences of R2R3-MYB TFs, we randomly chosed PsMYB114L, PsMYB12L and 16 other known MYB TFs related to anthocyanin biosynthesis from different species to perform the phylogenetic analysis. The functional role of the genes included in the analysis is known, and we have described it in the Figure 3. Furthermore, we have added additional interpretation on the results of the phylogenetic analysis to the revised manuscript (Lines 137-142).

13.Results: 2.2 Subcellular localisation. Also here a conclusion/comments would help. I would expect TFs to be nuclear localised. Is the localization found surprising? How does this relate to other localization of R2R3 MYB TFs?  Is it possible to include negative controls?

Response: Thank you very much for your suggestion. We have added some comments to the Subcellular Localization section of the revised manuscript (Lines 157-158; Lines 161-162). In terms of the localization of R2R3 MYB TFs, it is true that most of them were generally localized in the nucleus. However, based on many previous studies (Gong et al. Acta Hortic Sin. 2014, 41, 1400-1408; Ding et al. Acta Agric. Boreali-Occidentalis Sin. 2018, 27, 586-594, etc.), we found that it is possible for R2R3 MYB TFs or other TFs to simultaneously localize in the nucleus and cytoplasm. Furthermore, for the control, it was shown in the Figure 4A.

14.Results: 2.3 Expression analyses Line 140: “collected at 5 developmental stages” (à refer to  M&M and figure S2)

Response: Thank you very much for your suggestion. We have referred to Figure S2 and added the description of 5 developmental stages to the revised manuscript (Line 174; Lines 182-184). 

15.Results: 2.3 Expression analyses In figure 5 put the expression of PsMYB114L and PsMYB12L above and then the graphs of the other genes below (as described in the text). Increase font size in the graphs as it is hardly possible to read.

Response: Thank you very much for your suggestion. We have put the expression of PsMYB114L and PsMYB12L above and the graphs of the other genes below in the revised manuscript. Furthermore, we have increased the font size in the graphs (Figure 5)

16.Results: 2.3 Expression analyses Line 143: this is the first time genes like FLS and ANR are mentioned (a part from the abstract). Their known role should be explained in the introduction.

Response: Thank you very much for your suggestion. For FLS and ANR genes mentioned for the first time, we have added explanations to the Introduction section of the revised manuscript (Lines 69-72).

17.Results:2.4 overexpression in Arabidopsis. Expression of candidate genes in transgenics Arabidopsis should be performed and compared with the expression analyses done in Apple calli to have stronger indications of the reliability of the results obtained in the latter.

Response: Thank you very much for your suggestion. We have added the expression ananlysis of candidate genes in transgenic Arabidopsis in Section 2.4 of the Results section of the revised manuscript (Lines 193-204; Lines 210-212).

18.Results:2.4 overexpression in Arabidopsis. The CDS of the 2 TFs were isolated in Paeonia (which tissue?). Their expression has been tested in different stages of the petals. What is the phenotype of Arabidopsis flowers? Do they accumulate anthocyanins and do they present a coloured phenotype?

Response: The CDS of the 2 TFs were isolated from the petals of P. suffruticosa ‘Shima Nishiki’ (Lines 372 and 395). As for the phenotype of Arabidopsis flowers, compared with Col-0, it still presented a white color phenotype in the transgenic lines.

19.Results: 2.4 overexpression in Arabidopsis. Line 152 to 158 belongs to material and methods section.

Response: Thank you very much for your suggestion. we have moved Lines 152 to 158 to the Materials and Methods section of the revised manuscript (Lines 398-404).

20.Results: 2.5 overexpression in Apple calli. explain the choice of apple calli system (could Paeonia´s calli be an option?). Explain why “light and low temperature treatments were performed” (plus explain in the M&M the details of these treatments)

Response: At present, the genetic transformation system of apple calli is relatively well known, and it has been widely used in transgenic studies related to anthocyanins. In contrast, the genetic transformation system of Paeonia calli is still relatively unknown. To date, no transgenic studies related to anthocyanins have been reported using Paeonia calli. Therefore, we choose to use apple calli for transgenic studies. By referring to previous transgenic studies related to anthocyanins (An, J.P. Hortic. Res. 2017, 4, 17023; An, J.P. Plant Cell Physiol. 2017, 58, 1953-1962), we performed light and low temperature treatments for phenotypic observation. We have explained the details of these treatments in the Materials and Methods section of the revised manuscript (Lines 426-427).

21.Discussion: Integrate and elaborate the results from 2.3 with the results from 2.4 (and of the Arabidopsis qPCR I requested above). e.g. in 2.3 114L peaks in stage 3 as FLS and ANR suggesting perhaps a positive correlation between the genes. However in 2.4 upon overexpression of 114L, FLS and ANR are strongly downregulated. The authors should try to interpret these and other results.

Response: Thank you very much for your suggestion. We have added the expression ananlysis of candidate genes in transgenic Arabidopsis of PsMYB114L and PsMYB12L in Section 2.4 of the Results section of the revised manuscript. Furthermore, by comparing and analyzing these qRT-PCR results in Paenonia suffruticosa,  Arabidopsis, and apple calli, we found that there were differences to a certain extent in the expression patterns of some anthocyanin biosynthesis-related genes (e.g. FLS, DFR, etc.). We considered that this expression difference is likely caused by the promoter sequence specificity of this gene in different plant species. We have added the relevant interpretation to the Discussion section of the revised manuscript (Lines 294-300; Lines 316-320).

22.Discussion: Does it make sense that most structural genes for anthocyanin biosynthesis are downregulated in 35S:114L but still these calli produce more anthcyanins? Authors should comment and interpret these results, surely also by adding more background information of the genes analysed.

Response: Among the downregulated structural genes, the expression levels of MdFLS and MdANR (especially MdFLS) genes in 35S:114L were significantly lower than those in the WT. Furthermore, the expression levels of MdDFR and MdANS genes in 35S:114L were higher than those in the WT. Furthermore, it is true that the competition between FLS and DFR enzyme activities regulates different branches of the flavonoid biosynthesis pathways. Therefore, the significant downregulation of the FLS gene and the upregulation of the DFR gene would activate the branch of the anthocyanin biosynthesis, resulting in anthocyanin accumulation. The details of the relevant interpretation have been described in the Discussion section of the revised manuscript (Lines 301-303; Lines 306-307).

23.Discussion: Line 218: change “and” with “whereas” or “while” to show contrasts. bHLH-interaction motif ([D/E]Lx2[R/K]x3Lx6Lx3R) existed in the R3 domain of PsMYB114L, whereas PsMYB12L did not contain this motif for interaction with bHLH proteins.

Response: Thank you very much for your suggestion. We have changed “and” with “whereas” in the revised manuscript (Line 263).

24.Discussion: Line 233: remove “,respectively” otherwise it seems that 114L was overexpressed in Arabidopsis (only) and 12L in apple calli (only)

Response: Thank you very much for your suggestion. We have removed the word “,respectively” in the revised manuscript (Line 279).

25.Materials and Methods: It is unclear how the genes were selected, the primers designed and genes amplified. In materials and methods they said they have the genome? and how the names were given

Response: We apologize that this information was not clear in the original manuscript. As suggested, we have added more detail regarding these aspects to the revised manuscript (Lines 98-101; Lines 103-106; Lines 369-371; Lines 381-382; Lines 450 and 452).

26.Materials and Methods: From which tissue and conditions was obtained the cDNA from which the 2 TFs were amplified and cloned

Response: We apologize that this information was not clear in the original manuscript. The cDNA of PsMYB114L/PsMYB12L gene amplification and clone were obtained from the petals of P. suffruticosa ‘Shima Nishiki’ under field conditions. The details have been described in the revised manuscript (Lines 372 and 395).

27.Materials and Methods: The description of the phylogenetic analyses is missing

Response: We apologize that this information was not clear in the original manuscript. We have added a description of the phylogenetic analyses to the revised manuscript (Lines 450-452).

28.Conclusion: The first two sentences say exactly the same concept, remove one of the two.

Response: Thank you very much for your suggestion. We have removed the second sentence from the revised manuscript.

29.Conclusion: would then add some emphasis on the fact that in P. suffruticosa the two genes are detected in the petals (this manuscript-which might indicate a role in flower color) and, if more information is known from the published transcriptomic study (ref 50), mentioned it here to strengthen this hypothesis.

Response: Thank you very much for your suggestion. we have added some emphasis on the fact that in P. suffruticosa the two genes are detected in the petals to the revised manuscript (Lines 457-459).

30.English should be improved. I will point out some examples along the manuscript but I suggest a careful a thorough editing work.

Response: Thank you very much for your suggestion. We have carefully checked the language used and the sentence structure of the revised manuscript. Additionally, a certifi­cate of English language editing is now provided as supplementary material (Figure S4).

Round 2

Reviewer 2 Report

I aknowledge the authors for carefully editing the manuscript and answering my criticisms and comments. I hope this has helped improving the manuscript. I have a few additional remarks which I have included in yellow highlight in their responses: My references to lines are according to the manuscript with track changes.

1. The authors revised lines 17 and 18 but that does not correspond to what mentioned here and it still does not make sense to me from a language point of view.

8. I appreciated the inclusion of the diagram in fig1

9. Thanks, this makes it cleared to the reader. However, I would still add the reference of the transcriptomic publication.

10. Thanks for adding the deposited sequence numbers. However, I would place the accession numbers in the material and methods, where in general these information belong.

15. Thanks for the effort to make Figure 5 more comprehensible. However, I think that the font and quality is still not quite readable. I hope it will be better in the published version.

17. Thanks for including the expression analysis in transgenic Arabidopsis. There is a mistake in line 212 “both of the MdDFR/MdANS genes showed” but here it is not about Malus domestica genes. Should be AtDFR/AtANS.

21. Thanks for the effort to integrate qPCR data from several experiments. Unfortunately I do not see yet an analysis of the data in figure 5, which in this way remains rather scopeless.

English was certainly improved. In addition to what I have pointed out above I found two additiona spelling mistakes:

Line 269-“ceratin R2R3-MYB TFs regulate anthocyanin production in P. suffruticosa” Ceratin should be certain

Line 296: I think should be “the  leaves of transgenic Arabidopsis plants” rather than “their”

Finally, although it is certainly good to have gone through some English language editing, I have never seen the certificate being added as (supplementary) figure to a scientific paper. Unless it is common practice for IJMS which I ignore, I would suggest removing the certificate from the published article.

Author Response

Dear reviewer, 

Thank you very much for your comments on our manuscript entitled “Identification of two novel R2R3-MYB transcription factors, PsMYB114L and PsMYB12L, related to anthocyanin biosynthesis in Paeonia suffruticosa” (ijms-446453). We appreciate the comments from you and have made sincere efforts to address them in the revised manuscript. The manuscript was carefully revised according to the comments and suggestions, and our point-by-point responses to your comments are listed below. The line numbers refer to those in the revised manuscript without track changes. Furthermore, we have used a differently colored font for all changes in the revised manuscript with track changes (blue words or sentences with strikeout indicate that the words or sentences were deleted, and red words or sentences indicate that the words or sentences were changed or added).

We sincerely hope that the revised manuscript meets the publication standards for International Journal of Molecular Sciences.

1.Abstract: The authors revised lines 17 and 18 but that does not correspond to what mentioned here and it still does not make sense to me from a language point of view.

Response: We apologize that this was not clear in the first revised manuscript. We have modified this phrasing “Flower color is a charming phenotype phenotypic characteristic with considerable ornamental and commercial value” to “Flower color is a charming phenotype with very important ornamental and commercial values” in the second revised manuscript (Lines 17-18).

2.Results: 2.1: Thanks, this makes it cleared to the reader. However, I would still add the reference of the transcriptomic publication.

Response: Thank you very much for your suggestion. We have added the reference [39] of the transcriptomic publication in the second revised manuscript (Line 98).

3.Results: 2.1: Thanks for adding the deposited sequence numbers. However, I would place the accession numbers in the material and methods, where in general these information belong.

Response: Thank you very much for your suggestion. We have placed the accession numbers in the Material and Methods section of the second revised manuscript (Lines 379-380).

4.Results: 2.3: Thanks for the effort to make Figure 5 more comprehensible. However, I think that the font and quality is still not quite readable. I hope it will be better in the published version.

Response: Thank you very much for your suggestion. We have made the font and quality of Figure 5 better in the second revised manuscript.

5.Results:2.4: Thanks for including the expression analysis in transgenic Arabidopsis. There is a mistake in line 212 “both of the MdDFR/MdANS genes showed” but here it is not about Malus domestica genes. Should be AtDFR/AtANS.

Response: We apologize for this writing mistake. We have modified the phrase “MdDFR/MdANS” to the phrase “AtDFR/AtANS” in the second revised manuscript (Line 197).

6.Discussion: Thanks for the effort to integrate qPCR data from several experiments. Unfortunately I do not see yet an analysis of the data in figure 5, which in this way remains rather scopeless.

Response: We apologize for the absence of the analysis of the data in figure 5. We have added the relevant interpretation of this data in figure 5 to the Discussion section of the second revised manuscript (Lines 293-296 and Lines 317-325).

7. English was certainly improved. In addition to what I have pointed out above I found two additiona spelling mistakes:

Line 269-“ceratin R2R3-MYB TFs regulate anthocyanin production in P. suffruticosa” Ceratin should be certain

Line 296: I think should be “the  leaves of transgenic Arabidopsis plants” rather than “their”

Response: We apologize for these spelling mistakes. We have replaced the words “ceratin” and “their” with the words “certain” and “the” in the second revised manuscript (Lines 252 and 279).

8.Finally, although it is certainly good to have gone through some English language editing, I have never seen the certificate being added as (supplementary) figure to a scientific paper. Unless it is common practice for IJMS which I ignore, I would suggest removing the certificate from the published article.

Response: Thank you very much for your suggestion. we have removed the certificate from the Supplementary Material section.